# Bayesian Knowledge Distillation for Online Action Detection

## Abstract

Online action detection aims at identifying the ongoing action in a streaming video without seeing the future. Timely and accurate response is critical for real-world applications. In this paper, we introduce Bayesian knowledge distillation (BKD), an efficient and generalizable framework for online action detection. Specifically, we adopt a teacher-student architecture. During the training, the teacher model is built with a Bayesian neural network to output both the feature mutual information that measures the informativeness of historical features to ongoing action and the detection uncertainty. For efficient online detection, we also introduce a student model based on the evidential neural network that learns the feature mutual information and predictive uncertainties from the teacher model. In this way, the student model can not only select important features and make fast inference, but also efficiently quantify the prediction uncertainty by a single forward pass. We evaluated our proposed method on three benchmark datasets including THUMOS'14, TVSeries, and HDD. Our method achieves competitive performance with much better computational efficiency and much less model complexity. We also demonstrate that BKD generalizes better and is more data-efficient by extensive ablation studies. Finally, we validate the uncertainty quantification of the student model by performing abnormal action detection.

## 1 Introduction

Online action detection (OAD) De Geest et al. (2016) aims at identifying the ongoing action in a streaming video based on the historical observations. Different from the offline setting Shou et al. (2016); Zeng et al. (2019), it only has the past information and makes the prediction as soon as the action takes place. It has many important applications such as autonomous driving Chen et al. (2020), visual surveillance Sultani et al. (2018), and human-robot interaction Goodrich et al. (2008). OAD is challenging due to the incomplete observations of actions and redundant information among inputs such as background and irrelevant actions. Also, computational efficiency is a big concern for development on edge devices with limited computing resources. In addition, capturing the predictive uncertainty and generalizing to unseen environments are difficult, which are required by safety-critical applications such as autonomous driving.

To address these challenges, we introduce Bayesian knowledge distillation (BKD) for efficient and generalizable online action detection. We aim to learn a lightweight model that can make fast inference and uncertainty quantification. Specifically, we adopt a teacher-student architecture Hinton et al. (2015) for knowledge distillation. The teacher model is built in a Bayesian manner to model the posterior distribution of model parameters. It employs the mutual information (MI) between historical features and the ongoing action to select the most informative features for current action detection. The teacher model also quantifies the epistemic uncertainty of the action detection. The teacher model, however, is computationally expensive and is thus not suitable for online action detection. To address this challenge and inspired by evidential deep learning Ulmer et al. (2023) for uncertainty quantification, we proposed an evidential probabilistic student model. It is composed of an attention network and an evidential probabilistic neural network. The mutual information and distribution of the teacher model are distilled to these two components respectively. In this way, the student model can generate spatial-temporal attention masks that select important features with high mutual information. In the meantime, it can also quantify the predictive uncertainty by a single

forward pass since it inherits the knowledge of the Bayesian teacher model. As the knowledge of Bayesian teacher model is transferred, we name this process as Bayesian knowledge distillation.

We evaluate the performance and efficiency of BKD on benchmark datasets including THU-MOS'14 Idrees et al. (2017), TVSeries De Geest et al. (2016), and HDD Ramanishka et al. (2018). We also demonstrate the data-efficiency and generalization ability of BKD by experiments with reduced training data and cross settings. In the end, we validate the uncertainty quantification of student model by performing and abnormal action detection using the quantified uncertainties.

**Contributions.** 1) We introduce a Bayesian deep learning model for active online action detection that selects the most informative features based on mutual information and quantifies the action detection uncertainties. 2) we introduce an evidential deep learning model that quantifies its detection uncertainties in a single forward pass. We further introduce a knowledge distillation procedure that distills mutual information and predictive uncertainties from the Bayesian model to the evidential model, allowing the evidential network to perform feature selection using mutual information and to output Bayesian predictive uncertainties. After the distillation, the evidential model can efficiently select the most informative features and to quantify predictive uncertainty. 3) Our proposed BKD achieves competitive performance on benchmark datasets with faster inference speed and less model complexity. We also demonstrate it generalizes better and is more data-efficient.

## 2 RELATED WORK

**Online action detection.** For the model architecture, RNN-based designs Li et al. (2016); Gao et al. (2017); De Geest & Tuytelaars (2018); Xu et al. (2019); Eun et al. (2020; 2021); Chen et al. (2022); Han & Tan (2022); Kim et al. (2022b); Gao et al. (2021); Ye et al. (2022) are widely adopted because of RNN's temporal modeling capability. Typically, Xu Xu et al. (2019) proposed temporal recurrent network (TRN) that leverages both the historical information and predicted future features to detect the ongoing action. Thanks to the self-attention mechanism and the parallel computing property, Transformer-based methods Wang et al. (2021); Xu et al. (2021); Yang et al. (2022); Kim et al. (2022a); Zhao & Krähenbühl (2022); Rangrej et al. (2023); Hedegaard et al. (2022); Cao et al. (2023); Wang et al. (2023) become the mainstream for online action detection. Wang Wang et al. (2021) proposed OadTR that makes use of both historical information and future prediction. Xu Wang et al. (2021) proposed long short-term Transformer (LSTR) that captures both the long-range and short-term dependencies by two memory units. To overcome the latency of feature extraction, Cao et al. (2023) proposed E2E-LOAD for end-to-end online action detection. Besides RNN and Transformer, graph modeling is also studied for online action detection Elahi & Yang (2022). To leverage the video-level annotations instead of the dense frame-level annotations, weakly-supervised methods detection Gao et al. (2021); Ye et al. (2022) are also explored for OAD.

**Knowledge distillation.** Recently, deep neural networks have been widely applied for real-world applications. However, the scale of the model and computation cost are increasing dramatically, which raises challenges to devices with limited resources Gou et al. (2021). Thus, knowledge distillation (KD) has been introduced for model compression and acceleration. Here we review the main KD techniques based on teacher-student architecture, which is adopted by our BKD. Firstly, some approaches revise the teacher model with fewer layers and fewer channels in each layer Wang et al. (2018); Zhu et al. (2018); Li et al. (2020). In another way, the quantized version of the teacher model is saved to be used as the student model Polino et al. (2018); Mishra & Marr (2017); Wei et al. (2018). Also, there are work building small networks with efficient basic operations Howard et al. (2017); Zhang et al. (2018); Huang et al. (2017). Besides, global network structure can also be optimized to build the student model Liu et al. (2020); Xie et al. (2020); Gu & Tresp (2020).

**Evidential deep learning for uncertainty estimation.** Different from existing Bayesian uncertainty estimation methods Lakshminarayanan et al. (2017); Gal & Ghahramani (2016); Lakshminarayanan et al. (2017) that use multiple parameter sets and forward passes, evidential deep learning aims at using a factorization of the posterior predictive distribution to allow computing uncertainty in a single forward pass and with a single set of weights Ulmer et al. (2023). Specifically, it postulate that the target variable adheres to a conjugate distribution, whose parameters are treated as random variables. Some approaches Malinin & Gales (2018; 2019); Nandy et al. (2020) acquire conjugate distributions from out-of-distribution (OOD) data. Other methods Charpentier et al. (2020); Chen et al. (2018); Sensoy et al. (2018) leverage ensemble knowledge to learn the conjugate distribution.

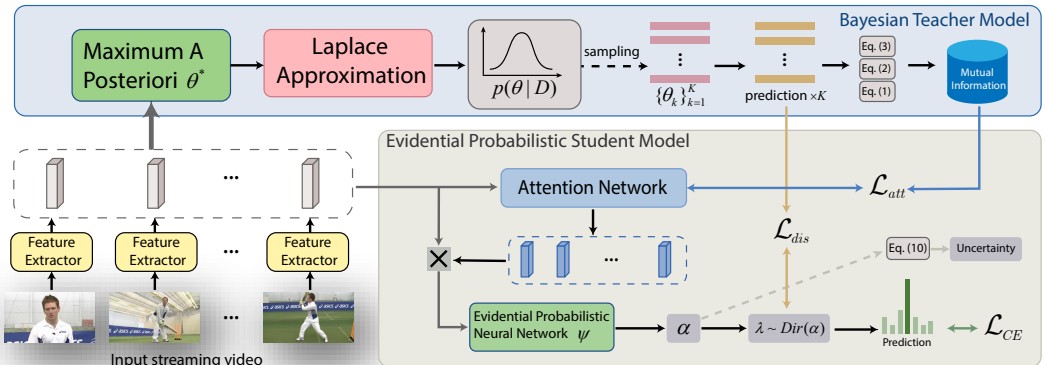

Figure 1: **Overall framework of Bayesian knowledge distillation (BKD).** The input of model is a streaming video. A pretrained backbone is used to extract the features of each frame. During the training, the Bayesian teacher model generates the mutual information (MI) and multiple sets of predictions. The student model is composed of an attention network and an evidential probabilistic neural network, which learn the MI and distributions from the teacher model respectively. It can actively select relevant features to perform fast inference and uncertainty quantification.

## 3 PROPOSED APPROACH

In this section, we first give an overview of the proposed Bayesian knowledge distillation (§ 3.1). Then we formulate the online action detection task (§3.2). Next, we introduce the Bayesian teacher model (§3.3) and evidential probabilistic student model (§3.4). In the end, we give the training and inference procedures (§3.5).

### 3.1 OVERVIEW

The overall framework is shown in Figure 1. The input is a streaming video, features are extracted by a pretrained backbone. During the training, the Bayesian teacher model generates the mutual information and action predictions, which are distilled to the evidential probabilistic student model for feature selection and uncertainty quantification respectively. During the testing, only the student model is kept to perform fast inference and efficient uncertainty quantification.

### 3.2 PROBLEM FORMULATION

**Online action detection(OAD)** aims at recognizing the ongoing action in a streaming video with only the past and current observations. Denote the input video as $\boldsymbol{V} = [I_1, I_2, ..., I_T]$, where $T$ is the length of video and $I_t$ denotes the frame at current time $t$. The online action detection is formulated as a classification problem: $y_t^* = \arg\max_c p(\hat{y}_t = c | \boldsymbol{V}_t)$, where $\hat{y}_t$ is the prediction, $c$ is the class label, and $\boldsymbol{V}_t = \{I_1, ...I_t\}$ is the available frame set at time $t$. A feature extractor is used to process each frame and generate the corresponding feature vector. Denote the feature set at time $t$ as $\boldsymbol{F}^t = \{F_1^t, ..., F_t^t\}$. The feature at time $i$ is $F_i^t \in \mathbb{R}^J$, where $J$ is the feature dimension of each frame.

### 3.3 BAYESIAN TEACHER MODEL

One of the objectives of the teacher model is to generate the mutual information (MI) between past features and the ongoing action. The MI indicates the relevance of features so it can be used to supervise the student model for feature selection. Denote a past feature as $F_{ij}^t$, where $i \in \{1, ..., t\}$ is the time index, $j \in \{1, ..., J\}$ is the feature index within each frame, and $t$ denotes the current time. We aim to obtain the mutual information between $F_{ij}^t$ and the ongoing action $y_t$. An illustration is shown in Figure 2. To compute mutual information, we build the teacher model in a Bayesian manner. Different from point estimation, Bayesian method constructs a posterior distribution of model parameters. By integrating predictions from multiple models, it is less likely to be overfitting and the predictive uncertainty can be accurately quantified. Additionally, Bayesian method is more robust

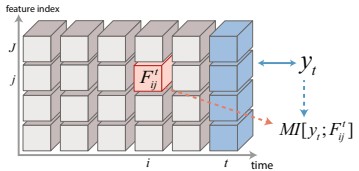
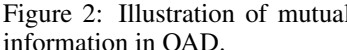

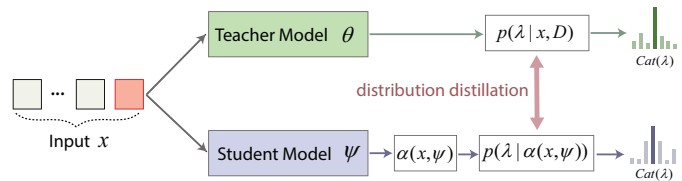

Figure 2: Illustration of mutual information in OAD.

Figure 3: Illustration of distribution distillation by evidential probabilistic neural network.

when training data is insufficient. We term the mutual information computed using the Bayesian method Bayesian Mutual Information (BMI).

Denote the model parameters of the teacher model as $\theta$ and we treat them as probability distributions. Then the BMI between a past feature $F_{ij}^t$ and the ongoing action $y_t$ can be written as:

$$\mathcal{I}[y_t; F_{ij}^t|\mathcal{D}] = \mathcal{H}[y_t|\boldsymbol{F}_{-ij}^t, \mathcal{D}] - \mathcal{H}[y_t|\boldsymbol{F}^t, \mathcal{D}] \tag{1}$$

where $\mathcal{D}$ denotes the training data, $\mathcal{H}$ denotes the entropy, and $\boldsymbol{F}_{-ij}^t$ is the feature set at time $t$ excluding $F_{ij}^t$, i.e. $\boldsymbol{F}_{-ij}^t = \boldsymbol{F}^t/F_{ij}^t$. By definition, the entropy term is written as:

$$\mathcal{H}[y_t|\boldsymbol{F}^t, \mathcal{D}] = -\sum_{y_t \in \mathcal{Y}} p(y_t|\boldsymbol{F}^t, \mathcal{D}) \log p(y_t|\boldsymbol{F}^t, \mathcal{D}) \tag{2}$$

where $\mathcal{Y} = \{0, 1, ..., C\}$ is the action class set. 0 represents background class and $C$ is the number of action classes. The prediction distribution in Eq. (2) can be computed as:

$$p(y_t|\boldsymbol{F}^t, \mathcal{D}) = \int p(y_t|\boldsymbol{F}^t, \theta)p(\theta|\mathcal{D})d\theta \approx \frac{1}{K}p(y_t|\boldsymbol{F}^t, \theta_k), \text{ where } \theta_k \sim p(\theta|\mathcal{D}) \tag{3}$$

In Eq. (3), we use the sample average to approximate it since it is impractical to integrate over all the possible parameters. Similarly, we can compute $\mathcal{H}[y_t|\boldsymbol{F}_{-ij}^t, \mathcal{D}]$.

To obtain the posterior distribution $p(\theta|\mathcal{D})$, we perform the Laplace approximation (LA). To reduce the computation cost, we adopt a last-layer Bayesian setting Kristiadi et al. (2020). Only the posterior distribution of last-layer parameters is modeled, while keeping the remaining parameters deterministic. Later, we will show this is efficient and effective. Specifically, we assume the last-layer parameters follows a Gaussian distribution.

Firstly, we train the teacher model under the deterministic setting by a cross entropy loss:

$$\theta^* = \arg\min_\theta \mathcal{L}(\mathcal{D}; \theta) = \arg\min_\theta \left( \sum_{n=1}^N l(x_n, y_n; \theta) + r(\theta) \right)$$
$$= \arg\min_\theta \left( \sum_{n=1}^N -\log p(y_n|f_\theta(x_n)) - \log p(\theta) \right) \tag{4}$$

where $\mathcal{D} = \{x_n, y_n\}_{n=1}^N$ is the training set and $r(\theta)$ is a regularizer such as weigh regularizer (a.k.a. weight decay). So $\theta^*$ is indeed a maximum a posteriori (MAP) estimate.

The Laplace approximation uses a second-order Taylor expansion of $\mathcal{L}(\mathcal{D}; \theta)$ around $\theta^*$ to construct a Gaussian approximation to $p(\theta|\mathcal{D})$:

$$\mathcal{L}(\mathcal{D}; \theta) \approx \mathcal{L}(\mathcal{D}; \theta^*) + \frac{1}{2}(\theta - \theta^*)^T (\nabla_\theta^2 \mathcal{L}(\mathcal{D}; \theta)|_{\theta^*})(\theta - \theta^*) \tag{5}$$

where the first-order term vanishes at $\theta^*$. So the Laplace posterior approximation can be obtained as:

$$p(\theta|\mathcal{D}) \approx \mathcal{N}(\theta^*, \Sigma), \text{ with } \Sigma := (\nabla_\theta^2 \mathcal{L}(\mathcal{D}; \theta)|_{\theta^*})^{-1} \tag{6}$$

After the LA, we sample $K$ times from $p(\theta|\mathcal{D})$ to obtain $K$ sets of parameters $\{\theta_1, ..., \theta_K\}$. Then we approximate the prediction distribution in Eq. (3) and further compute the mutual information of each feature. The BMI we computed are based on multiple models from the teacher model. We use BMI to supervise the attention network of student model.

### 3.4 Evidential probabilistic student model

While the teacher model can perform accurate feature selection, uncertainty quantification, and action detection, it is computationally expensive and hence is unsuitable for online action detection. To address this issue, we introduce a student model. The student model should be efficient and simultaneously inherits the Bayesian formalism in the teacher model. To achieve this goal, we propose to implement the student model using an evidential probabilistic neural network. Compared with teacher model, the student model is much smaller and can achieve real-time inference, while simultaneously employing mutual information to select features and to produce the predictive uncertainty.

**Attention network.** The attention network actively selects high mutual information features from the inputs by generating a spatial-temporal attention mask $\boldsymbol{A}_t$ using a fully-connected network and applying to the original features by element-wise product. In this way, irrelevant features are masked out since they have low mutual information to the ongoing action. The training of the attention network is supervised by the BMI $\mathcal{I}_t$ from the teacher model. By minimizing the mean squared error (MSE) loss $\mathcal{L}_{att}$ between $\boldsymbol{A}_t$ and $\mathcal{I}_t$, the attention network can generate the BMI-aware attention mask without computing the BMI. $\mathcal{L}_{att}$ can be written as below:

$$\mathcal{L}_{att} = MSE(\boldsymbol{A}_t, \sigma(\mathcal{I}_t)) = \frac{1}{tJ} \sum_{i,j} ||A_{ij}^t - \sigma(\mathcal{I}[y_t; F_{ij}^t | \mathcal{D}])||^2 \qquad (7)$$

where $\sigma(\cdot)$ is the sigmoid function.

**Evidential probabilistic neural network (EPNN) and distribution distillation.** To distill the knowledge and enable the student model to efficiently quantify predictive uncertainty, we introduce EPNN. We aim to transfer the distribution of the Bayesian teacher model to the student model, namely the distribution distillation. By doing so, the student model can capture the predictive uncertainty since it inherits the knowledge of Bayesian teacher model.

Under the classification setting, the target output of student model $y$ follows a categorical distribution with parameter $\lambda$, $y \sim p(y|\lambda) = Cat(\lambda)$. For example, $\lambda$ represents the probability after the final softmax layer. We treat $\lambda$ as a random variable and assume it follows Dirichlet distribution, i.e. $\lambda \sim p(\lambda|\alpha(x, \psi)) = Dir(\alpha(x, \theta))$, where $\alpha$ denotes the parameters of the Dirichlet distribution and $\psi$ denotes student model parameters. Similarly, the teacher model has posterior distribution $p(\lambda|x, \mathcal{D}) = \int p(\lambda|x, \theta) p(\theta|\mathcal{D}) d\theta$. During the distillation, we transfer teacher posterior distribution $p(\lambda|x, \mathcal{D})$ to the student posterior distribution $p(\lambda|\alpha(x, \psi))$. An illustration is shown in Figure 3. Specifically, we minimize the KL-divergence between these two distributions as below:

$$\begin{aligned} \mathcal{L}_{dis} &= KL(p(\lambda|x, \mathcal{D}) || p(\lambda|\alpha(x, \psi))) \\ &\propto -\sum_{c=1}^{C} \log(\Gamma(\alpha_c)) + \log \Gamma(\sum_{c=1}^{C} \alpha_c) - \mathbb{E}_{p(\theta|\mathcal{D})}[\sum_{c=1}^{C} (\alpha_c - 1) \log \lambda_c(x, \theta)] \end{aligned} \qquad (8)$$

where $C$ is the number of action classes. Detailed derivation of $\mathcal{L}_{dis}$ can be found in Appendix. Since the different predictions of the teacher model. The complete distribution distillation procedures are summarized in Algorithm 1 in the Appendix.

**Joint training.** The student model is jointly trained for online action detection, attention distillation, and distribution distillation. The total loss function $\mathcal{L}$ is below:

$$\mathcal{L} = \mathcal{L}_{ce} + \lambda_1 \mathcal{L}_{att} + \lambda_2 \mathcal{L}_{dis} \qquad (9)$$

where $\mathcal{L}_{ce}$ is the cross-entropy loss for online action detection. $\lambda_1$ and $\lambda_2$ are hyper-parameters that emphasize the attention network and distribution distillation. Although the distribution distillation can make the student model perform online action detection, we still perform joint training because the Bayesian teacher model is not perfect and its posterior distribution is approximated, which may introduce error to the student model. The joint training brings two benefits: 1) the online action can be improved with the supervision of ground-truth label; 2) the negative log-likelihood $\mathcal{L}_{ce}$ makes the training faster and more stable as indicated in Fathullah et al. (2023).

**Uncertainty quantification.** After training the EPNN, the Dirichlet distribution of the student model obtained the knowledge of the Bayesian teacher model. The uncertainties can be computed by a single forward pass with closed-form solutions. Apart from the total predictive uncertainty, the

student model can also output the epistemic uncertainty, which indicates the limited knowledge in the modeling process.

Specifically, the total uncertainty and epistemic uncertainty can be quantified as:

$$\mathcal{H}[p(y|x,\theta)] = \sum_{c=1}^{C} \frac{\alpha_c}{\alpha_0} \log \frac{\alpha_c}{\alpha_0}, \text{ where } \alpha_0 = \sum_{c=1}^{C} \alpha_c$$

$$\mathcal{I}[y;\lambda|\alpha] = -\sum_{c=1}^{C} \frac{\alpha_c}{\alpha_0} \big( \ln \frac{\alpha_c}{\alpha_0} \Psi(\alpha_c + 1) + \Psi(\alpha_0 + 1) \big) \tag{10}$$

where $\Psi(\cdot)$ is the dgamma function. Details derivations can be found in Appendix.

### 3.5 Training and inference

To better train the student model, we adopt a two-stage training strategy. After obtain the mutual information from the teacher model, we first train the attention network of the student model with the attention loss $\mathcal{L}_{att}$ in Eq. 7. Then we jointly train the student model with the total loss $\mathcal{L}$ in Eq. 9. We demonstrate this works better in Sec. 4.5. During the inference, the student model only needs a forward pass to obtain the prediction and uncertainty quantification by Eq. 10.

## 4 Experiments

### 4.1 Datasets and evaluation metrics

**THUMOS'14** Idrees et al. (2017) is a dataset for video-based temporal action localization. We use the validation set with 200 videos for training and test set with 213 videos for evaluation. There are 20 action classes and a background class. We ignore the frames with ambiguous labels. Following the settings in Gao et al. (2017); Xu et al. (2019), we adopt the mean Average Precision (**mAP**) as the evaluation metric.

**TVSeries** De Geest et al. (2016) is a dataset collected from real TV series. It contains 27 episodes with 30 daily actions. It is a challenging dataset due to the viewpoints changing and occlusions in the videos. To counter the imbalanced data distribution, we adopt the mean calibrated Average Precision (**mcAP**) De Geest et al. (2016) as the evaluation metric. It is computed as $cAP = \sum_k cPrec(k) \times \mathbf{1}(k)/P$, where $cPrec = TP/(TP + FP/\omega)$, $P$ is the total number of positive frames and $\mathbf{1}(k)$ is an indicator function that is equal to 1 if frame $k$ is a true positive. The mcAP is the mean of calibrated average precision of all action classes.

**HDD** Ramanishka et al. (2018). HRI Driving Dataset (HDD) is a dataset for learning driver behavior in real-life environments. It contains 104 hours of real human driving in the San Francisco Bay Area collected by an instrumented vehicle with different sensors. There are 11 goal-oriented driving actions. Following the settings in Ramanishka et al. (2018), we use 100 sessions for training and 37 sessions for testing. And only the sensor data is used as the input. **mAP** is used as the evaluation metric for this dataset.

### 4.2 Settings

**Feature extraction.** Following the settings in Eun et al. (2020); Xu et al. (2019; 2021), we use TSN Wang et al. (2016) to extract the features for THUMOS'14 and TVSeries. Video frames are extracted at 24 fps and the chunk size is set to 6. To better capture the spatial-temporal dependencies, we adopt the multi-scale vision Transformer Fan et al. (2021) to extract RGB features. The optical flow features are extracted with BN-Inception Ioffe & Szegedy (2015). The backbone is pretrained on ActivityNet Heilbron et al. (2015) and Kinetics-400 Carreira & Zisserman (2017) separately for evaluation. For HDD dataset, the sensor data is used as the input.

**Implementation details.** The BKD framework is implemented in PyTorch Paszke et al. (2017). The model is trained by the Adam optimizer Kingma & Ba (2014) with a learning rate of $10^{-4}$ and a weight decay of $5 \times 10^{-5}$. The batch size is set to 32. The experiments were conducted on two

Table 1: **Experiment results on THUMOS'14, TVSeries and HDD.** The results on THUMOS'14 and HDD are reported as mAP (%). The results on TVSeries are reported as mcAP (%). For HDD, ⋆ indicates RGB data is used as the input.

| Method | Architecture | THUMOS'14 | | TVSeries | | HDD |
|---|---|---|---|---|---|---|
| | | ANet | Kinetics | ANet | Kinetics | Sensor |
| RED Gao et al. (2017) | RNN | 45.3 | - | 79.2 | - | 27.4 |
| FATS Kim et al. (2021) | | - | 59.0 | 81.7 | 84.6 | - |
| TRN Xu et al. (2019) | | 47.2 | 62.1 | 83.7 | 86.2 | 29.2 |
| IDN Eun et al. (2020) | | 50.0 | 60.3 | 84.7 | 86.1 | - |
| PKD Zhao et al. (2020) | | - | 64.5 | - | 86.4 | - |
| WOAD Gao et al. (2021) | | - | 67.1 | - | - | - |
| OadTR Wang et al. (2021) | Transformer | 58.3 | 65.2 | 85.4 | 87.2 | 29.8 |
| *Co*OadTR Hedegaard et al. (2022) | | 56.1 | 64.2 | 87.6 | 87.7 | 30.6 |
| Colar Yang et al. (2022) | | 59.4 | 66.9 | 86.0 | 88.1 | 30.6 |
| LSTR Xu et al. (2021) | | 65.3 | 69.5 | 88.1 | 89.1 | - |
| Uncertainty-OAD Guo et al. (2022) | | 66.0 | 69.9 | 88.3 | 89.3 | 30.1 |
| TeSTra Zhao & Krähenbühl (2022) | | 68.2 | 71.2 | - | - | - |
| GateHUB Chen et al. (2022) | | 69.1 | 70.7 | 88.4 | 89.6 | 32.1 |
| MAT Wang et al. (2023) | | **70.4** | 71.6 | **88.6** | 89.7 | **32.7** |
| E2E-LOAD Cao et al. (2023) | | - | **72.4** | - | **90.3** | 48.1⋆ |
| BKD (ours) | Transformer | 69.6 | 71.3 | 88.4 | 89.9 | 32.5 |

Nvidia RTX 3090 Ti GPUs. The number of epochs is set to 25. Ablation studies on existing methods are based on their officially released codes.

## 4.3 MAIN EXPERIMENT RESULTS AND COMPARISONS

We evaluate BKD on benchmark datasets and make a comprison with other methods in Table 1. The performance of BKD student model is reported. Our BKD achieves 71.3% mAP on THUMOS'14, which is slightly lower than the 72.4% from E2E-LOAD. On TVSeris, BKD achieves 89.9% mcAP, which is 0.4% lower than the state-of-the-art. On HDD, BKD achieves 32.5% mAP with sensor input, which is 0.2% lower than the MAT. In general, our BKD achieves very competitive performance for OAD. In the following parts, we show the other superior properties of BKD while keeping high detection accuracy.

Table 2: **Comparison of computation efficiency and model complexity.** The mAP is reported on THUMOS'14 with Kinetics-pretrained features. Our proposed BKD has less model complexity and much computational cost. And the inference speed is much faster than other methods.

| Method | Modality | Model | | Inference Speed (FPS) | | | | mAP (%) |
|---|---|---|---|---|---|---|---|---|
| | | Parameter Count | GFLOPs | Optical Flow Computation | RGB Feature Extraction | Flow Feature Extraction | Model | |
| TRN | RGB + Flow | 402.9M | 1.46 | 8.1 | 70.5 | 14.6 | 123.3 | 62.1 |
| OadTR | | 75.8M | 2.54 | 8.1 | 70.5 | 14.6 | 110.0 | 65.2 |
| LSTR | | 58.0M | 7.53 | 8.1 | 70.5 | 14.6 | 91.6 | 69.5 |
| GateHUB | | 45.2M | 6.98 | 8.1 | 70.5 | 14.6 | 71.2 | 70.7 |
| MAT | | 94.6M | - | 8.1 | 70.5 | 14.6 | 72.6 | **71.6** |
| BKD (ours) | | 18.7M | 0.46 | 8.1 | 70.5 | 14.6 | **169.6** | 71.3 |

## 4.4 COMPUTATIONAL EFFICIENCY AND MODEL COMPLEXITY

We build the Bayesian knowledge distillation in order to make it more practical for real-time applications. A comparison of computation efficiency and model complexity is shown in Table 2. While keeping a high detection accuracy, our proposed BKD has much less model complexity and computational cost. And the inference speed of the model is much higher compared to other methods. Recently, Cao et al. (2023) proposed an end-to-end OAD framework to avoid the latency from the frame and feature extraction parts, which leads to higher overall inference speed. Here, we mainly compare the model inference speed instead of the pre-processing part.

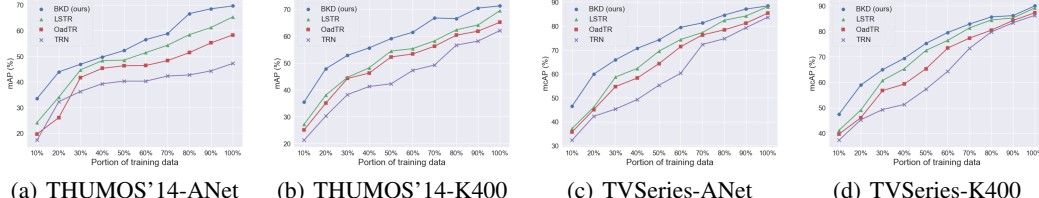

| (a) THUMOS'14-ANet | (b) THUMOS'14-K400 | (c) TVSeries-ANet | (d) TVSeries-K400 |

Figure 4: **Experiment results of training with small-scale data.** We reduce the training data from 100% to 10% and compared with TRN Xu et al. (2019), OadTR Wang et al. (2021), and LSTR Xu et al. (2021). The results are plotted for THUMOS'14 and TVSeries with both ActivityNet and Kinetics pretrained features. Our proposed BKD outperforms all others when training data is limited.

## 4.5 ABLATION STUDIES

**Training strategies.** To demonstrate the effectiveness of two-stage training strategy, we compared it with two more training strategies. One is jointly training the attention network and EPNN by the total loss in Eq. 9. The other is firstly training the attention network by the attention loss. Then we freeze the weights of attention network and train the EPNN by the $\mathcal{L}_{att}$ and $\mathcal{L}_{ce}$. We refer the first one as joint training and the second one as separate training. The performance comparison is shown in Table 3. The results show that the two-stage training outperforms the other two strategies on both THUMOS-14 and TVSeries with different features, which demonstrates its effectiveness.

**Training with small-scale data.** By applying the mutual information based feature selection mechanism, we expect BKD to be more data-efficient when the amount of training data is limited. We reduce the training data from 100% to 10% and compare with other methods. The results on THU-MOS'14 and TVSeries are plotted in Figure 4. When training data is reduced, BKD has less performance decay and outperforms other methods, which demonstrates that BKD is more data-efficient.

**Number of historical frames.** At time $t$, BKD takes a certain number of past frames to predict the ongoing action. To study the long-range and short-term modeling capability of BKD, we vary the number of past frames as the input. The experiment results are shown in Table 4. THUMOS'14 and TVSeries are extracted at 6 fps. HDD is extracted at 3fps. So the optimal number of frames on HDD is 32.

Table 3: **Ablation study of different training strategies.** The two-stage training gives the best performance.

| Training | THUMOS'14 | | TVSeries | |
|---|---|---|---|---|
| | ActivityNet | Kinetics | ActivityNet | Kinetics |
| Joint | 68.0 | 70.5 | 84.6 | 87.9 |
| Separate | 63.0 | 66.9 | 80.2 | 83.4 |
| Two-stage | **69.6** | **71.3** | **88.4** | **89.9** |

Table 4: **Ablation study of different number of past frames**

| Dataset | Feature | Number of past frames | | | | |
|---|---|---|---|---|---|---|
| | | 8 | 16 | 32 | 64 | 128 |
| THUMOS'14 | ANet | 37.3 | 57.3 | 68.0 | **69.6** | 65.9 |
| | Kinetics | 46.1 | 59.0 | 70.4 | **71.3** | 69.2 |
| TVSeries | ANet | 62.5 | 76.6 | 87.3 | **88.4** | 84.1 |
| | Kinetics | 66.5 | 78.9 | 88.4 | **89.9** | 87.9 |
| HDD | Sensor | 29.9 | 31.4 | **32.5** | 31.8 | 30.0 |

**Generalization.** To test the generalization capability of the model, we perform the Cross-View and under occlusion experiments on TVSeries dataset. Following the settings in Guo et al. (2022), the training set and test set are from different view angles. For the occlusion, the training data does not contain occlusion and the testing data is occluded. The experiment results and comparison are shown in Table 5. The results show that BKD generalizes better under different conditions.

**Balance of the loss function.** In the second training stage of BKD, the total loss in Eq. 9 is composed of three terms. $\lambda_1$ and $\lambda_2$ are the weights for attention loss and distribution distillation loss. To study the balance of all three terms, we train the model with different $\lambda$s on THUMOS-14 and TVSeries. the comparisons are plotted in Figure 5. We empirically choose $\lambda_1 = 0.4$ and $\lambda_2 = 6$ since they give the best performance.

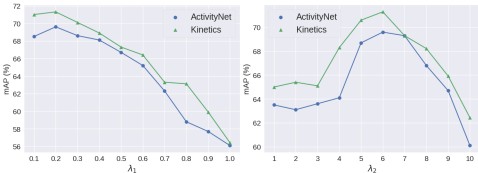

Figure 5: Ablation study of $\lambda_1$ and $\lambda_2$ on THU-MOS'14 with ActivityNet and K400 features.

Table 5: Cross-view and occlusion experiments results on TVSeries.

| Method | CV (%) | Occ. (%) |
|---|---|---|
| TRN Xu et al. (2019) | 65.8 | 85.2 |
| OadTR Wang et al. (2021) | 66.2 | 87.7 |
| Uncertainty-OAD Guo et al. (2022) | 67.3 | 89.5 |
| Colar Yang et al. (2022) | 66.7 | 88.3 |
| LSTR Xu et al. (2021) | 69.5 | 89.4 |
| TeSTra Zhao & Krähenbühl (2022) | 70.2 | 89.9 |
| BKD (ours) | 74.1 | 91.5 |

**Effectiveness of Bayesian mutual information.** To verify the effectiveness Bayesian mutual information, we first compare with the model without BMI. Specifically, we trained the student model with the same evidential probabilistic neural network without the attention network. The total loss function is $\mathcal{L} = \mathcal{L}_{ce} + \lambda_2 \mathcal{L}_{dis}$ and we grid-searched $\lambda_2$ to obtain the best accuracy. In addiction, we compute the mutual information based on a single prediction instead of BMI and trained the model by the same loss function in Eq.9. The results are shown in Table 6. From the results, the BKD with BMI outperforms other two methods, which demonstrates the effectiveness of the BMI.

Table 6: Effectiveness of BMI. BKD-No-BMI denotes the model without attention network and BKD-MI denotes the model with normal MI.

| Model | THUMOS'14 | | TVSeries | |
|---|---|---|---|---|
| | ActivityNet | Kinetics | AcivityNet | Kinetics |
| BKD-No-BMI | 63.1 | 64.8 | 86.5 | 87.4 |
| BKD-MI | 67.2 | 68.0 | 87.9 | 88.5 |
| BKD | **69.6** | **71.3** | **88.4** | **89.9** |

Table 7: Experiment results of abnormal action detection on THUMOS'14 and TVSeries.

| Uncertainty | THUMOS'14 (%) | TVSeries (%) |
|---|---|---|
| Total | 80.49 | 65.10 |
| Epistemic | 86.26 | 69.02 |
| Aleatoric | 62.51 | 39.83 |

### 4.6 VERIFICATION OF UNCERTAINTY QUANTIFICATION

**Abnormal action detection.** To verify the uncertainty quantification, we conduct experiments for abnormal action detection. In THUMOS'14, there are twenty action classes, a background class, and an "ambiguous" class. The ambiguous class refers to frames that are difficult to identify during the labeling process. We consider them as the outlier data or abnormal action. We utilize the quantified uncertainties in Eq. 10 as well as the aleatoric uncertainty to detect abnormal action. Specifically, if the predictive uncertainty is above a certain threshold, we declare it as an outlier. The experiment results on THUMOS'14 and TVSeries are shown in Table 7. From the results, the epistemic uncertainty leads to best detection accuracy and aleatoric uncertainty is the worst. This is because epistemic uncertainty captures the lack of knowledge and it is inversely proportional to the data density. And Aleatoric uncertainty captures the noise or randomness in the data. This verifies the effectiveness of the uncertainty quantification of the evidential probabilistic student model.

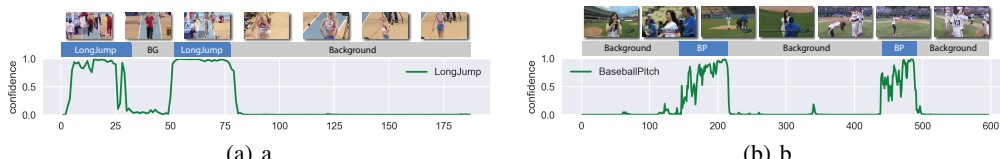

(a) a
(b) b

Figure 6: **Qualitative results.** The curves are the prediction confidence of the ground-truth actions.

## 5 CONCLUSION

We introduce Bayesian knowledge distillation for efficient and generalizable online action detection. By distilling the mutual information and distributions of a Bayesian teacher model to an evidential probabilistic student model. The student model can not only make fast and accurate inference, but also efficiently quantify the prediction uncertainty. The experiment results on benchmark datasets demonstrate the effectiveness of our proposed method for both OAD and uncertainty quantification.

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
