# BAYESIAN KNOWLEDGE DISTILLATION FOR ONLINE ACTION DETECTION

## A APPENDIX

### A.1 DERIVATION OF DISTILLATION LOSS FUNCTION

$$
\begin{aligned}
\mathcal{L}_{dis} &= KL(p(\lambda|x,\mathcal{D})||p(\lambda|\alpha(x,\psi))) \\
&\propto -\int p(\lambda|x,\mathcal{D})\log p(\lambda|\alpha(x,\psi))d\lambda \\
&= -\int\int p(\lambda|x,\theta)p(\theta|\mathcal{D})[\log p(\lambda|\alpha(x,\psi))]d\lambda d\theta \\
&= -\int p(\theta|D)[\log p(\lambda(x,\theta)|\alpha(x,\psi))]d\theta \\
&= -\sum_{c=1}^{C}\log(\Gamma(\alpha_c)) + \log\Gamma(\sum_{c=1}^{C}\alpha_c) - \mathbb{E}_{p(\theta|\mathcal{D})}[\sum_{c=1}^{C}(\alpha_c-1)\log\lambda_c(x,\theta)]
\end{aligned}
\tag{1}
$$

### A.2 COMPARISON OF FULL-BAYESIAN AND LAST-LAYER BAYESIAN FOR LAPLACE APPROXIMATION

For the teacher model, we adopt the last-layer Bayesian **?** method to reduce the difficulty of training. We also performed the Laplace approximation over all the model parameters. The comparison is shown in Table 1. The full-Bayesian method has consistent improvement on THUMOS-14 and TVSeries. But the LA in training takes much longer time than the last-layer Bayesian and the model needs careful tuning, so we adopt the last-layer Bayesian in the LA process.

| Method | THUMOS-14 (mAP %) | | TVSeries (mcAP %) | |
|---|---|---|---|---|
| | ActivityNet | Kinetics | AcivityNet | Kinetics |
| Full-Bayesian | 70.5.3 | 72.2 | 88.9 | 90.3 |
| Last-layer | 69.6 | 71.3 | 88.4 | 89.9 |

Table 1: **Comparison of full-Bayesian and last-layer Bayesian.** Full-Bayesian improves the performance since it models the distribution of all model parameters.

### A.3 FULL DISTRIBUTION DISTILLATION ALGORITHM

Here we summarize the full distribution distillation procedures in Algorithm 1, which including both teacher model training and student model training.

### A.4 EXPERIMENTAL RESULTS ON TVSERIES OF DIFFERENT PORTIONS OF VIDEOS

**Evaluation of different stages of action.** To evaluate the detection performance at different stages of the video, we show the results of using different portions of the video in Table 2 following the settings in **?**. The results show that BKD has consistent performance at different stages.

---

**Algorithm 1** Distribution distillation procedures

---

**Input:** $\mathcal{D} = \{(x_n, y_n)\}_{n=1}^{N}$ - Training data, where $x_n$ is the data sample, $y_n$ is the label, and $N$ is the total number of samples in the training set.

**Output:** $\theta_s$ - parameters of student model

    ***1 - Training teacher model***

    1.1 - Training of deterministic teacher model

1: Denote the parameters of the teacher model as $\Theta = \{\phi, \theta\}$, where $\phi$ includes the parameters before the last layer and $\theta$ includes the parameters of last layer.

2: **for** $n = 1$ to $N$ **do**

3:     Make the prediction of $x_n$ deterministically: $p(\hat{y}_n|x_n, \phi, \theta) \in \mathbb{R}^C$, where $C$ is the total number of action classes

4:     Compute the teacher cross-entropy loss $\mathcal{L}_{CE}^t = -\sum_{c=1}^{C} \mathbb{1}(y_n = c) \log p(\hat{y}_n = c|x_n, \phi, \theta) + r(\phi, \theta)$, where $r(\phi, \theta)$ is a regularizer (a.k.a. weight decay)

5:     Optimizing $\phi_t$ and $\theta$ by minimizing $\mathcal{L}_{CE}^t$

6: **end for**

7: Save the optimized model parameters $\phi^*$ and $\theta^*$

    1.2 - Laplace Approximation (last-layer)

8: Using LA technique to obtain $p(\theta|\mathcal{D}) \sim \mathcal{N}(\theta^*, -H^{-1})$,
where $\theta^*$ is the point-estimate obtained from the last step and $H = \nabla_{\theta_t}^2 \log p(\theta|\mathcal{D})|_{\theta=\theta^*}$ is the Hessian matrix

    1.3 - Computing mutual information

9: Compute the mutual information of each feature element following the procedures in Sec. 3.3

    ***2 - Distilling knowledge to student model***

    2.1 - Training of HPNN student model

10: Denote the parameters of the student model as $\psi$

11: **HPNN model**: $x \to \alpha \to \lambda \to y$, where $\lambda$ denotes the parameters of the categorical distribution $p(y|\lambda)$, and $\alpha$ is the parameters of distribution of $\lambda = Dir(\lambda|\alpha)$

12: **for** $n = 1$ to $N$ **do**

13:     Generate $\alpha_n \in \mathbb{R}^C$ by feeding $x_n$ into the model

14:     Sample $\lambda_n \in \mathbb{R}^C$ from $Dir(\lambda|\alpha_n)$

15:     Make the prediction of $x_n$ as $p(\hat{y}_n|\lambda_n)$

16:     Compute the distillation loss for HPNN: $\mathcal{L}_{dis} = KL[p(\lambda_n|x, \mathcal{D}, \phi^*)||p(\lambda_n|\alpha(x, \psi))] = \sum_{c=1}^{C} \log(\Gamma(\alpha_n^c)) + \log \Gamma(\sum_{c=1}^{C} \alpha_n^c) - E_{p(\theta|\mathcal{D},\phi^*)}[\sum_{c=1}^{C}(\alpha_n^c - 1) \log \lambda_n^c(x, \phi^*, \theta)]$

17:     Optimizing $\psi$ by minimizing $\mathcal{L}_{dis}$

18: **end for**

19: **return** Updated student model parameters $\psi$

---

Table 2: **Experimental results on TVSeries of different portions of videos in terms of mcAP (%).** Each portion is only used to compute mcAP after detecting the current actions on all frames in an online manner.

| Method | Feature | \multicolumn{10}{c}{Portion of video} | | | | | | | | | |
| --- | --- | --- | --- | --- | --- | --- | --- | --- | --- | --- | --- |
| | | 0-10% | 10-20% | 20-30% | 30-40% | 40-50% | 50-60% | 60-70% | 70-80% | 80-90% | 90-100% |
| TRN | | 78.8 | 79.6 | 80.4 | 81.0 | 81.6 | 81.9 | 82.3 | 82.7 | 82.9 | 83.3 |
| IDN | | 80.6 | 81.1 | 81.9 | 82.3 | 82.6 | 82.8 | 82.6 | 82.9 | 83.0 | 83.9 |
| OadTR | | 79.5 | 83.9 | 86.4 | 85.4 | 86.4 | 87.9 | 87.3 | 87.3 | 85.9 | 84.6 |
| Colar | ActivityNet | 80.2 | 84.4 | 87.1 | 85.8 | 86.9 | 88.5 | 88.1 | 87.1 | 86.6 | 85.1 |
| TFN | | 83.1 | 84.4 | 85.4 | 85.8 | 87.1 | 88.4 | 87.6 | 87.0 | 86.7 | 85.6 |
| LSTR | | 83.6 | 85.0 | 86.3 | 87.0 | 87.8 | 88.5 | 88.6 | 88.9 | 89.0 | 88.9 |
| BKD (ours) | | **84.9** | **86.3** | **86.9** | **87.8** | **88.7** | **88.9** | **90.4** | **90.6** | **90.3** | **90.3** |
| IDN | | 81.7 | 81.9 | 83.1 | 82.9 | 83.2 | 83.2 | 83.2 | 83.0 | 83.3 | 86.6 |
| PKD | | 82.1 | 83.5 | 86.1 | 87.2 | 88.3 | 88.4 | 89.0 | 88.7 | 88.9 | 87.7 |
| OadTR | Kinetics | 81.2 | 84.9 | 87.4 | 87.7 | 88.2 | 89.9 | 88.9 | 88.8 | 87.6 | 86.7 |
| LSTR | | 84.4 | 85.6 | 87.2 | 87.8 | 88.8 | 89.4 | 89.6 | 89.9 | 90.0 | 90.1 |
| GateHUB | | **84.5** | 87.6 | 89.5 | 90.0 | 90.2 | 91.0 | 91.3 | 91.3 | 91.3 | 90.7 |
| BKD (ours) | | 84.4 | **87.9** | **88.6** | **90.7** | **91.5** | **91.9** | **92.0** | **92.0** | **91.6** | **91.5** |