# OpenReview forum: "Bayesian Knowledge Distillation for Online Action Detection"
_ICLR.cc/2024/Conference — Submitted to ICLR 2024_

### Official Review · Reviewer_kgXL · 2023-10-25

**Soundness:** 3 good
**Presentation:** 3 good
**Contribution:** 2 fair
**Rating:** 5
**Confidence:** 4

**Summary:**

This paper proposed a method based on Bayesian knowledge distillation for online action detection. A teacher-student framework is proposed. By distilling the mutual information and distributions of a Bayesian teacher model to an evidential probabilistic student model. The student model can not only make fast and accurate inference, but also efficiently quantify the prediction uncertainty. Experimental results demonstrate the effectiveness of the proposed method.

**Strengths:**

Pros:
1. A method based on Bayesian knowledge distillation is proposed, which makes inference more efficient.

2. Experimental results verify the effectiveness of the proposed method.

**Weaknesses:**

Cons:
1. This paper seems to simply combine existing knowledge distillation and uncertainty techniques. Knowledge distillation is already proposed in PKD (Zhao et al. (2020)), and uncertainty technique have been used in Uncertainty-OAD (Guo et al. (2022)) for online action detection.

2. More details about the teacher model should be included. The results of the teacher model are also missing.

3. More visual analysis should be included instead of all numerical analysis.

4. Can the performance of the student model boost by increasing the number of model parameters? I wonder when the performance of the student model can exceed that of the existing state-of-the-art methods when the number of parameters is increased.

**Questions:**

See Weaknesses for more details, and limitations should also be included.

---

> ### Author Response · Authors · 2023-11-21
> **Answers to reviewer kgXL**
>
> Thanks for your detailed comments, here are our answers to your concerns:
>
> 1) Th knowledge distillation in PKD (Zhao et al. (2020)) aims at distilling the knowledge of offline model to an online model. Our knowledge distillation aims at distilling the distribution and mutual information of an online teacher model to an online student model and improving the efficiency. The overall mechanism is different. In addition, our student model can quantify the predictive uncertainty in a single forward pass, which is practical for real applications.
> 2) The teacher model is trained in a deterministic way. The Laplace approximation is performed as a post-processing. Code will be made publicly available. For the teacher model, we include the performance below:
>
> | Dataset | ActivityNet (mAP %) | Kinetics (mcAP%) |
> |-----------|------|--------|
> |THUMOS'14|71.6|73.9|
> |TVSeries|88.9|90.6|
>
> The teacher model has bettern performance than the student model and SOTAs. But it is more computationally expensive, which does not satisfies the objective of our overall goal
> 3) We will update the paper with visual analysis. Demos will also be available on our project page
> 4) By increasing the number of self-attention layers, our proposed BKD can achieve 72.3% mAP (SOTA: 71.6 %) on THUMOS’14 with 26.9M parameters (SOTA: 94.6M). Will update the results and efficiency in the newer version.

---

> > ### Comment · Reviewer_kgXL · 2023-11-23
> >
> > Thanks to the authors for the response. Despite there are some differences between the proposed method and existing techniques, I still view that the contribution is marginal and cannot meet the high standards of ICLR. Therefore, I will maintain my score rate and hope the authors to revise the manuscript according to the reviewers' comments.

---

### Official Review · Reviewer_i6Lf · 2023-10-31

**Soundness:** 2 fair
**Presentation:** 3 good
**Contribution:** 2 fair
**Rating:** 3
**Confidence:** 4

**Summary:**

This paper explores a Bayesian knowledge distillation (BKD) framework for the online action detection task, which aims at identifying the ongoing action in a streaming video without seeing the future. Specifically, the teacher model is a Bayesian neural network which outputs the mutual information between historical features and ongoing action and predicts the detection uncertainty. The student model learns useful information from the teacher model and constructs a Dirichlet distribution for uncertainty quantification. Competitive performances are achieved on multiple benchmarks.

**Strengths:**

+ As far as I am concerned, this is the first work which applies distillation on the online action detection task, which may provide some inspiration for the community.

**Weaknesses:**

- Lack of novelty. Although it may be the first work which applies the distillation architecture on online action detection, there is little innovation on components design and theoretical analysis. Teacher-student architecture, Bayesian neural network, and attention network are all very common tools.
- Misleading/Inappropriate use of evidential deep learning (EDL). EDL is based on the Subjective Logic theory, which is implemented by its unique optimization objective and is accompanied by its own uncertainty calculation method. In the work, the authors construct a simple Dirichlet distribution and then claim they adopt EDL, which is not true.
- Unfair (or at least incomplete) comparison of computation efficiency and model complexity. Authors claim that the proposed method BKD achieves competitive performance with less model complexity and computational cost, and provide comparison results on Table 2. However, the other methods did not adopt a teacher-student distillation manner as BKD, and the Bayesian neural network which is used as the teacher model by BKD is quite computationally heavy. It is a very natural result for BKD to achieve fast inference at the cost of much larger computation in the training phase via distillation, and the comparison of training speed is not provided.
- Careless writing. For example, there are two very obvious citation mistakes on the first page of supplementary material.

**Questions:**

1. More discussion about the unique novelty of this work may be provided.
2. Why do the authors use CE loss and Eq.(10) for model optimization and uncertainty quantification, instead of using the EDL loss and the EDL uncertainty estimation method?
3. A comparison of training time is necessary for the completeness of experiments.
For others please refer to the Weakness.

---

> ### Author Response · Authors · 2023-11-21
> **Answers to reviewer i6Lf**
>
> Thanks for your insightful comments, here are our answers to your concerns:
>
> 1) The novelties of this paper mainly lie in: (1) we are the first work leveraging knowledge distillation to improve the inference efficiency of online action detection, which is a big concern for real applications. (2) The student model are trained to identify important features automatically based on the mutual information. (3) We build an evidential student network, which can quantify the predictive uncertainty by a single forward pass.
>
> 2) The concept of evidential deep learning is first introduced by [1] to quantify uncertainty for classification problems. By applying a Dirichlet distribution to class probabilities, they treated predictions of a neural network as subjective opinions and learned the function to collect the evidence leading to these opinions from data. The uncertainty estimation is guided by the theory of evidence. Subsequent research has aimed to enhance uncertainty quantification in the framework of [1], particularly by including out-of-distribution data or density models. These developments are comprehensively reviewed in the survey paper [2], which also inspired our use of the term “evidential deep learning”. For a more detailed discussion on this topic, please refer to [2]. We will further clarify evidential deep learning for quantifying classification uncertainty.  \
> [1] Sensoy, Murat, Lance Kaplan, and Melih Kandemir. "Evidential deep learning to quantify classification uncertainty." Advances in neural information processing systems 31 (2018).\
> [2] Ulmer, Dennis, Christian Hardmeier, and Jes Frellsen. "Prior and posterior networks: A survey on evidential deep learning methods for uncertainty estimation." Transactions on Machine Learning Research (2023).
>
> 3) We agree the training of Bayesian neural network is expensive. Our proposed BKD performs Laplace approximation (LA) after training one deterministic teacher model. So the main computation gain lies in the LA process. Following the comparison in E2E-LOAD, we made a comparison of training cost with other methods below, the mAP is reported on THUMOS’14 with Kinetics-pretrained features :
> | Method | mAP (%) | GPU Mem | Time (min/epoch) | # of parameters |
> |----------|------|-------------------------------|--------|-----|
> | LSTR | 59.2 | 8x31.4 |1.5 | 105.9|
> | E2E-LOAD | 72.4 | 8x16.9 |9.6 | 53.5|
> | BKD (ours) | 71.3 | 2x24 |13.2 | 18.7|
>
> In addition, the inference speed is the first priority of online action detection so the inference FPS is the most important factor.
> For the unfair comparison concern, the knowledge distillation is one of our contributions. If other methods also leverage knowledge distillation, the performance will drop as the student model usually performs worth than the teacher model.
>
> 4) Thanks for your careful reading and pointing out the writing errors, we will proofread the paper and fix all of them.
>
>
> Question of using CE loss:
> Our work focuses on evidential deep learning that is designed specifically for classification uncertainty, as detailed in Section 3 of [2]. We use the Dirichlet distribution to efficiently capture uncertainty, allowing it to be determined in a single forward pass of the neural network. This is because Dirichlet distribution is conjugate to the target categorical distribution of y. Since traditional EDL's uncertainty quantification might not effectively distinguish between aleatoric and epistemic uncertainties, we address this through knowledge distillation to minimize the distance between the Dirichlet distribution and the corresponding distribution of the teacher model. By distilling both epistemic and aleatoric information from the Bayesian teacher model, we can employ Eq. 10 as outlined in Section 3.3 of [2] to separate these two types of uncertainty.

---

> > ### Comment · Reviewer_i6Lf · 2023-11-22
> > **Thank you for your response**
> >
> > Thank you for the response. However, some issues are still not addressed. For example, the relation between the proposed method and EDL; and the computational burden of the proposed method.
> > Besides, I also agree with the opinion of Reviewer 37DN.
> > Therefore, I prefer to maintain my original rating.

---

### Official Review · Reviewer_37DN · 2023-11-01

**Soundness:** 2 fair
**Presentation:** 2 fair
**Contribution:** 2 fair
**Rating:** 5
**Confidence:** 2

**Summary:**

This paper tried to handle a practical video detetion problem, online action detection without seeing the future frames. The authors introduced Bayesian knowledge distillation as a teacher network and evidential probabilistic neural network as a student network. The proposed method is evaluated on three benchmark datasets including THUMOS’14, TVSeries, and HDD and shows the efficiency. Ablation studies are conducted to prove the efficiency of the proposed method.

**Strengths:**

Online action detection aims at identifying the ongoing action in a streaming video without seeing the future and it is a practical problem for the video analysis. The authors tried to resolve a real problem with the motivation behind, to make the whole network inference efficient. They tried the teacher and student architecture and introduced Bayesian knowledge distillation as a teacher network and evidential probabilistic neural network as a student network. Comprehensive experimental results on several benchmark datasets are provided.

**Weaknesses:**

The paper appears to be more of an attempt at work rather than containing a substantial amount of insights or analysis. The paper introduced the Bayesian knowledge distillation, (for the first time in knowledge distillation?), however the authors did not provide much insights, such as why it will make the learning efficient etc. The same issue happend to the student network, the authors just introduced the network into the paper without much explainations. From my point of view, I did not understand why this paper should stand out due to two important proposals. Meanwhile, the experimental results shown in the paper are pretty cherry picked, such as MAT (Wang et.al 2023) is compared in different tables. However, in the Figure 4, the authors did not show MAT, on the contrary, the authors showed papers published before 2022.

The performance in the paper is not impressive, whatever in mAP or FPS.

**Questions:**

The questions are listed in the weakness section. Please address these questions.

---

> ### Author Response · Authors · 2023-11-22
> **Answers to reviewer 37DN**
>
> Thanks for you detailed and insightful comments, here are our answers to your concerns:
> 1) We did not introduce Bayesian knowledge distillation for the first time. It was initially introduced by Hinton et al. [1], who transferred knowledge from complex ensemble models to a single deterministic model. This idea was further developed by Korattikara et al. [2], who applied it to distill knowledge from Bayesian neural networks into deterministic networks. However, our work incorporates Bayesian knowledge distillation into online action recognition for the first time. While [1] and [2] focus only on distilling the prediction mean from a Bayesian model, our method also distills the uncertainty from the teacher model, focusing on the entire prediction distribution rather than just the mean.
> 2) Efficiency concern: We would like to clarify that our approach enhances inference efficiency, not necessarily learning efficiency. The student model, designed to be much smaller than the complex teacher model, ensures efficient inference for predictions. Additionally, the student model shows improved efficiency in uncertainty quantification. A key advantage of our student model is its ability to capture uncertainty, which is distilled from the teacher model during the knowledge distillation process. By employing evidential deep learning, the student model can calculate uncertainty with just a single forward pass, leveraging the Dirichlet distribution's conjugacy with the target categorical distribution of y. This contrasts with the teacher model, which is less efficient in both prediction and uncertainty estimation.
> 3) For the comparison with MAT (Wang et.al 2023), we only incorporate the results reported in the paper. We will reproduce their results and include them in the ablation study part.
> 4) For the performance concerns, our proposed BKD achieves very close performance compared to SOTAs on all benchmarks. In Table 2 of the paper, BKD achieves 169.6 FPS with 18.7M parameters, which is 46.3 FPS faster than the second fast TRN. Compared to MAT (72.6 FPS, 94.6M parameters), BKD has 0.3% lower performance drop but it has huge FPS improvement and parameters space saving.  \
> \
> [1] Hinton, Geoffrey, Oriol Vinyals, and Jeff Dean. "Distilling the knowledge in a neural network." arXiv preprint arXiv:1503.02531 (2015). \
> [2] Korattikara Balan, Anoop, et al. "Bayesian dark knowledge." Advances in neural information processing systems 28 (2015).

---

> > ### Comment · Reviewer_37DN · 2023-11-22
> >
> > Thanks for your replies. However, to be honest, I still think this paper is not above the bar of acceptance. To be, "Incorporates Bayesian knowledge distillation into online action recognition for the first time" is not the contributions or shows any novelty. I hope authors could provide more insights, such as why Bayesian knowledge distillation for the action recognition is critical. I still think this paper is pretty engineering.

---

### Official Review · Reviewer_f1zi · 2023-11-02

**Soundness:** 2 fair
**Presentation:** 2 fair
**Contribution:** 2 fair
**Rating:** 3
**Confidence:** 4

**Summary:**

In this paper, the authors present Bayesian knowledge distillation (BKD), a framework for online action detection that is both efficient and generalizable. The authors utilize a teacher-student architecture to improve efficiency. A key aspect of the proposed method is the introduction of a student model based on the evidential neural network. This student model learns feature mutual information and predictive uncertainties from the teacher model. With this design, the student model can not only select important features and make fast inferences, but also accurately quantify prediction uncertainty with a single forward pass. The proposed method was evaluated on three benchmark datasets: THUMOS’14, TVSeries, and HDD.

**Strengths:**

1. Scope: Online action detection is a crucial task for various applications, including autonomous driving, visual surveillance, and human-robot interaction. This paper addresses challenges such as incomplete action observations and computational efficiency. The focus of the paper is how a model can generalize to unseen environments. Overall, the work is relevant to the ICLR community.
2. The authors propose Bayesian knowledge distillation (BKD) as a solution for efficient and generalizable online action detection. They utilize a teacher-student architecture for knowledge distillation and leverage the student model to enable efficient inference.

**Weaknesses:**

1. Contribution #1: The authors assert that the proposed Bayesian deep learning model contributes to the task of active online action detection. However, the reviewer has concerns regarding this claim for the following reasons:
    a. Firstly, the reviewer agrees that inference speed is an important consideration and acknowledges the adoption of a teacher-student model. However, the authors have not provided justification for the limitations of existing teacher-student architectures for online action detection. It remains unclear why the authors chose to leverage evidential deep learning for this purpose. Additionally, there is no comparison of different teacher-student architectures, making it difficult for the reviewer to understand the rationale behind this design choice.
    b. Secondly, the motivation behind incorporating uncertainty prediction into the design is unclear. While the reviewer acknowledges the need to consider uncertainty due to the inherent unpredictability of the future, the authors have not highlighted the limitations of not modeling uncertainty. For example, a deterministic prediction of the current action may not be necessary, and predictions of potential opportunities for different actions may suffice.
    c. The authors present an experiment in Figure 6 to demonstrate the use of uncertainty quantification for abnormal action detection, specifically in the context of THUMOS. However, the experiment appears to be relatively simple and may be considered cherry-picked. A comprehensive evaluation of the proposed framework is necessary to validate its effectiveness.
2. The second claim is that the proposed framework can perform feature selection using mutual information and output Bayesian predictive uncertainties. The reviewer was expecting experimental evidence to support this claim. However, the reviewer did not find sufficient evidence regarding the effectiveness of the architectural design. To substantiate this claim, the reviewer requests experiments that can validate its effectiveness.
3. Experiments: As mentioned in point 1 (a), the authors primarily focus on demonstrating that the proposed framework performs on par with existing methods on benchmarks for online action detection. However, the reviewer believes that there is a lack of insights regarding the ablation of teacher-student architectures. To establish the value of the proposed evidential deep learning approach, the reviewer requests experiments that can validate this claim.

**Questions:**

The reviewer identified several major concerns in the Weakness section and would like to know the authors' thoughts on these points. Please answer each concern in the rebuttal stage. The reviewer will respond according to the authors' rebuttal in the discussion phase.

---

> ### Author Response · Authors · 2023-11-22
> **Answers to reviewer f1zi**
>
> Thanks for your insightful comments and  questions. Here are our answers to your concerns:
> 1) Weakness 1a: Based on our knowledge, we are the first work to adopt knowledge distillation to improve the efficiency of online action detection. (PKD [1] uses knowledge distillation to transfer the knowledge of an offline model to the online model, which is a different mechanism compared to our work). Traditional knowledge distillation methods typically involve single deterministic models for both teacher and student, aiming to align the student's predictions with those of the teacher. However, our approach employs Bayesian knowledge distillation. This technique not only distills the teacher model's predictions but also its associated uncertainty, focusing on the entire prediction distribution rather than just the mean. We leverage evidential deep learning for its efficacy in Bayesian knowledge distillation, particularly for uncertainty quantification efficiency. By transferring the teacher model's knowledge to a Dirichlet-based network, the student model attains not only accurate predictions but also efficient uncertainty estimation. The student model's uncertainty calculation requires just a single forward pass of the network, thanks to the Dirichlet distribution's conjugacy with the target categorical distribution of y. In contrast, the teacher model is less efficient in both prediction and uncertainty estimation. \
> \
> Weakness 1b: The student model learns from the teacher model. We aim to transfer as much as the knowledge to the student model, i.e. we aim to distill the distribution of the teacher model to student model. In addition, online action detection is used for many safety-critical appplications such as autonomous drving. Therefore, the uncertainty quantification is important for decision making which another motivation of modeling the uncertainty.  \
> \
> Weakness 1c: In Table 7 of the paper, we used different types of uncertainties for abnormal action detection. Since the abnormal action in our experiment setting denotes the classes that are not used for training, the epistemic uncertainty gives the best detection performance, which demonstrates the correctness of our uncertainty quantification.
>
> 2) To validate that the proposed framework can leverage mutual information to select important features, we re-implemented the framework without the attention module. The other parts of the framework remain the same. The experiment results are shown as follows:
>
> | Method | THUMOS'14-ANet | THUMOS'14-Kinetics | TVSeries-ANet | TVSeries-Kinetics |
> |----------|------|-------------------------------|--------|-----|
> | BKD (no attention) | 68.2 | 70.1 | 88.0 | 88.9 |
> | BKD | 69.6 | 71.3 | 88.4 | 89.9|
>
> The results demonstrate the effectiveness of our proposed attention modeling with mutual information.  \
>
> For the effectiveness of predictive uncertainty quantification, the results in Table 7 of the paper show that the uncertainties can be used for detecting abnormal actions accurately.
>
> 3) For the insights of teacher-student architectures. We implemented different knowledge distillation methods with a same teacher model for comparison. We did grid searches to select the optimal hyper-parameters for fair comparisons. The experiment results are shown as follows:
> | Method | THUMOS'14-ANet | THUMOS'14-Kinetics | TVSeries-ANet | TVSeries-Kinetics |
> |----------|------|-------------------------------|--------|-----|
> | PKD [1] | - | 64.5 | - | 86.4 |
> | Feature-based [2] | 65.2 | 68.1 | 87.0 | 88.4 |
> | Output-based [3] | 66.1 | 68.3 | 87.4 | 88.5 |
> | BKD | 69.6 | 71.3 | 88.4 | 89.9|
>
> The results shown our BKD obtains better performance than other knowledge distillation methods, which demonstrates the effectiveness of Bayesian distillation.  \
> \
> [1] Zhao, Peisen, et al. "Privileged knowledge distillation for online action detection." arXiv preprint arXiv:2011.09158 (2020). \
> [2] Hinton, Geoffrey, Oriol Vinyals, and Jeff Dean. "Distilling the knowledge in a neural network." arXiv preprint arXiv:1503.02531 (2015). \
> [3] Romero, Adriana, et al. "Fitnets: Hints for thin deep nets." arXiv preprint arXiv:1412.6550 (2014).

---

> > ### Comment · Reviewer_f1zi · 2023-11-23
> > **Thanks to the authors for the response.**
> >
> > Thanks to the authors for the response. Thanks for conducting additional results.
> >
> > 1. The response did not answer the question of why we need Bayesian formulation for action recognition. The authors focus on sharing the proposed design, instead of limitations of existing methods.
> >
> > 2. Thanks for sharing additional results. However, the improvements are marginal, which makes it difficult to convince others of the value of the proposed module.
> >
> > 3. There are "new" teacher-student frameworks. It is not convincing to convince the audience with the framework proposed in 2014 and 2015.

---

### Meta-Review · Area_Chair_XoWL · 2023-12-09

**Metareview:**

This paper addresses online action detection with a teacher-student framework using Bayesian knowledge distillation.

The reviewers cite problems with clarity, in explaining the motivation, lack of experimental details and support, computational requirements, etc.  All four reviewers feel the paper is not yet ready for acceptance and the AC concurs.

**Justification For Why Not Higher Score:**

The reviewers cite problems with clarity, in explaining the motivation, lack of experimental details and support, computational requirements, etc.  All four reviewers feel the paper is not yet ready for acceptance.

**Justification For Why Not Lower Score:**

N/A

---

### Decision · Program_Chairs · 2024-01-16

Reject